# Seismic Response of Flat Ground and Slope Models through 1 *g* Shaking Table Tests and Numerical Analysis

**Yong Jin [1], Hoyeon Kim [1], Daehyeon Kim [1,\*] , Yonghee Lee [2] and Haksung Kim [2]**

[1] Department of Civil Engineering, Chosun University, Gwangju 61452, Korea; a695023492@vip.qq.com (Y.J.); dizlks8542@chosun.kr (H.K.)

[2] Research Engineer Site & Structural Engineering Group Plant Construction & Engineering Lab. KHNP Central Research Institute, Daejeon 305-343, Korea; dragon202@khnp.co.kr (Y.L.); haksung.kim@khnp.co.kr (H.K.)

\* Correspondence: dkimgeo@chosun.ac.kr; Tel.: +82-062-230-7607

**Abstract:** In order to verify the reliability of numerical analysis, a series of 1 *g* shaking table tests for flat ground and slope were conducted using a laminar shear box subjected to different seismic waves. Firstly, numerical analyses, using the DEEPSOIL and ABAQUS software, were done to compare the results of flat ground experiments. After that, finite element analyses with ABAQUS were conducted to compare the results of slope experiments. For numerical analyses, considering the influence of the boundary, the concept of adjusted elastic modulus was proposed to improve the simulation results. Based on the analyses, it is found that in terms of acceleration-time history and spectral acceleration, the numerical analysis results are in good agreement with the experiment results. This implies that numerical analysis can capture the dynamic behavior of soil under 1 *g* shaking table test conditions.

**Keywords:** numerical analysis; 1 *g* shaking table test; elastic modulus; laminar shear box; boundary; acceleration-time history; spectral acceleration





## 1. Introduction

Due to the randomness and uncertainty of earthquakes, it is necessary to conduct simulated earthquake experiments to understand the dynamic behavior of soil in controlled environments. Generally, theoretical analysis, model test, and numerical analysis are the three major research methods for seismic responses. As a number of theoretical and numerical analysis results have not been validated due to the complexity of soil, a controlled model test such as the 1 *g* shaking table test is very useful.

Typically, 1 *g* shaking table tests with either rigid soil boxes or laminar shear boxes have been widely used to understand the dynamic response of soil subjected to earthquake loading. Rigid soil boxes have generally been used as they are very simple to use [1]. Laminar shear boxes have been recently used by some researchers for 1 *g* shaking table tests. Many researchers have attempted to obtain the seismic response as accurately as possible under 1 *g* shaking table test conditions. Therefore, the design of the model box is very important. The laminar shear box is successful in simulating the soil boundary conditions [2]. Turan et al. [3] studied the performance of the laminar box and non-linear seismic behavior of the model clay. The study showed that the laminar box does not impose significant boundary effects and is able to maintain 1-D soil column behavior. In addition, the dynamic behavior of the model clay during scaled model tests was found to be consistent with the behavior measured during cyclic laboratory tests. Chen et al [2] developed a laminar shear container and tested the natural vibration frequency and damping ratio of a model soil box. The results showed that the developed laminar shear soil container was successful in simulating the soil boundary conditions. It is an ideal for the container to weaken the reflection and scattering effect of waves on the boundaries.

Lee et al. [4] presented the boundary effects introduced by a laminar container for simulating 1D shear wave propagation in a free field, based on dynamic centrifuge tests. A series of 1D centrifuge shaking table tests involving dry and saturated sand models were conducted to simulate level deposits subjected to several magnitudes of 1 Hz base shaking. The time histories of acceleration and excess pore water pressure (saturated sand model) were measured at various depths and at several distances from the end walls to assess and quantify the boundary effects of the laminar container on the seismic response.

Since the 1980s, the rapid development of computer technology has greatly promoted the development of numerical solutions for ground seismic response analysis. The whole ground system can be calculated dynamically by the finite element method. Many scholars have done finite element analyses to evaluate the seismic response of soil [5–8]. Andersen [5] presented a numerical model for studying the dynamic evolution of landslides and analyzed a simplified slope with houses placed on the top. Faris and Wang [6] performed finite element calculation of seismic acceleration in shear zone of landslide using ABAQUS 2D model. Cheng et al. [7] studied the seismic response characteristics of saturated soft free field ground by a large-scale shaking table test. The nonlinearity coupled numerical model of dynamical effective stress of saturated soft free foundation was established using OpenSEES. Moghadam and Baziar [8] established a numerical model of the effect of a circular subway tunnel on the acceleration at the ground surface through 1 $g$ shaking table test to study the influence of soil shear wave velocity, input motion frequency content, flexibility ratio and depth of the tunnel on the amplification pattern. Previous studies have used single numerical analysis method, which may lead to a lack of reference in data comparison. This study uses two numerical analysis methods to compare with experiments to make the results more convincing. In general, it is known that the 1 $g$ shaking table test has some limitations for scale effects. For this reason, most studies have typically presented 1 $g$ shaking table test results qualitatively without thorough comparison with numerical results. The major purpose of the study is to compare the 1 $g$ shaking table test results with the numerical results in a very sound way, leading to the enhanced reliability of the 1 $g$ shaking table test results.

In this paper, numerical analysis and 1 $g$ shaking table test were done to evaluate dynamic behavior of soil. Numerical analyses using the DEEPSOIL and ABAQUS software were performed to model the flat ground and slope. The laminar shear box was used to simulate the flat ground and slope. The accuracy of numerical analysis and 1 $g$ shaking table test was verified by comparing the 6 numerical and experimental results.

## 2. Materials and Methods

### 2.1. Soil Properties

The soil sample of the scaled-down model test in this study was collected from a cut slope at a construction site located at Ulju-gun, in the Ulsan Metropolitan City area in Korea.

Table 1 shows the geotechnical index properties of the specimen used in this study. Specific gravity test, grain size test, standard proctor test and relative density test and physical properties of the soil were analyzed. The specific gravity of the soil was 2.69, the maximum dry unit weight was 18.27 kN/m$^3$, and the minimum dry weight was 12.43 kN/m$^3$. The optimum moisture content was 12.5% and the Atterberg limit test showed Non-Plastic (NP) for Plastic Index (PI). The maximum and minimum void ratios were 1.123 and 0.443, respectively. The fines content of the soil was 10.8% and the soil was classified as SW-SM according to the Unified Soil Classification System. For the 1 $g$ shaking table test, the specimen was selected for the sample passing through the No. 4 sieve after the physical property tests, and the sample remaining in the No. 4 sieve was about 1%. The elastic modulus was $2 \times 10^8$ Pa. According to the existing experimental data, the friction angle of 30° and the dilatancy angle of 24.4° were used in the study [9].

**Table 1.** Geotechnical index properties of the specimen used in this study.

| Parameter | Value | Parameter | Value |
|---|---|---|---|
| No.200 Passing (%) | 10.8 | $e_{max}$ | 1.123 |
| Gs | 2.69 | $e_{min}$ | 0.443 |
| OMC (%) | 12.5 | $r_{d\,max}$ $(kN/m^3)$ | 18.27 |
| PI (%) | NP | $r_{d\,min}$ $(kN/m^3)$ | 12.43 |
| USCS | SW-SM | Elastic modulus (Pa) | $2 \times 10^8$ |
| Internal friction angle (°) | 27.7° | Dilatancy angle (°) | 24.4° |

The elastic modulus is obtained indirectly using experimental data. The calculation of the modulus of elasticity is explained here. The average shear wave velocity was determined to be 72 m/s by combining the hammer test and the calculation formula of Hardin and Richart [10]. In a homogeneous and isotropic soil, the speed of the shear wave is controlled by the elastic modulus, as shown in Equation (1):

$$G = \rho V_s{}^2 \tag{1}$$

where G is the shear modulus, $\rho$ is the unit weight of soil, $V_s$ is the shear wave velocity.

The relationship between elastic modulus and shear modulus is:

$$G = \frac{E}{2(1+\nu)} \tag{2}$$

where, G is the shear modulus, E is the elastic modulus, $\nu$ is the Poisson's ratio.

According to Equations (1) and (2), knowing the unit weight of soil, $\rho$ is $1.47 \times 10^5$ N/m$^3$ and shear wave velocity, $V_s$ is 72 m/s. The shear modulus can be obtained as $7.6 \times 10^7$ Pa. Then knowing the shear modulus, G is $7.6 \times 10^7$ Pa, and the Poisson's ratio, $\nu$ is equal to 0.3, the elastic modulus can be obtained as $2 \times 10^8$ Pa. This modulus was used in the study. In order to make the parameters fit better, the concept of adjusted elastic modulus is also used later.

### 2.2. 1 g Shaking Table Test Equipment

The experimental equipment is mainly composed of a 1 *g* shaking table test equipment system, laminar shear box and accelerometers.

### 2.2.1. 1 g Shaking Table Test Equipment System

Figure 1 presents a schematic of the 1 *g* shaking table test system, which is mainly composed of a model, a hydraulic actuator rail, a base, a hydraulic actuator, a hydraulic pump unit, and a console. The hydraulic pump unit is an analog controller with an electro-hydraulic servo valve as the core, and its performance affects the entire system. It plays a decisive role and is the core part of the entire control system. The hydraulic actuator mainly provides power, including hydraulic pump stations, accumulator groups and cooling systems, etc. The rail of hydraulic actuator is designed according to the maximum speed value of the seismic wave. To save energy, large-capacity accumulator groups are currently installed to provide the actuators with huge instantaneous energy.

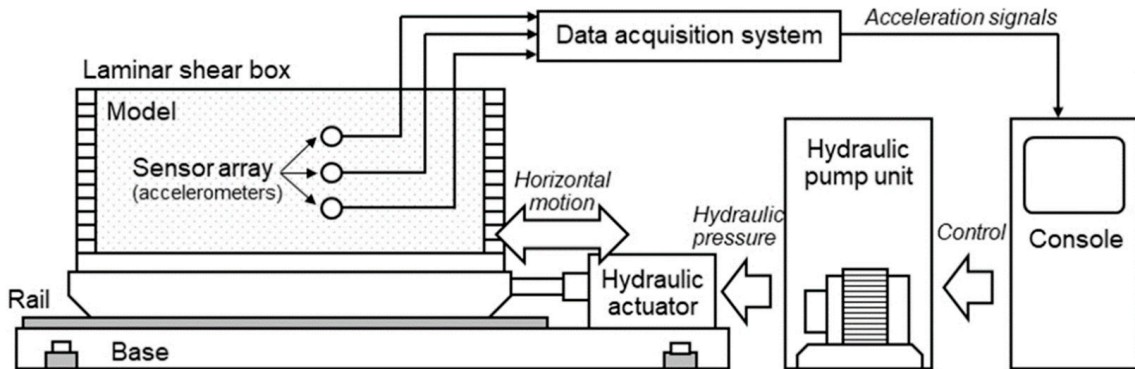

**Figure 1.** The experimental system used in this study (Kim et al, [11]).

### 2.2.2. Laminar Shear Box

Figure 2 shows the flexible soil box, referred to as a laminar shear box (LSB), used in this study. The dimensions of the LSB are 200 cm (horizontal), 60 cm (vertical) and 60 cm (height), with a total of 12 layers of aluminum frame structure. The thickness of each layer is 4.5 cm, and the spacing between each layer is about 0.5 cm. Each layer can move independently in a horizontal direction through a rolling bearing. The wall is made of aluminum. The main amplification period of the box is measured by resonance method with sinusoidal input continuous sweep. The natural period of the empty box and the box filled with soil is 0.04–0.05 s and about 0.1 s, respectively.

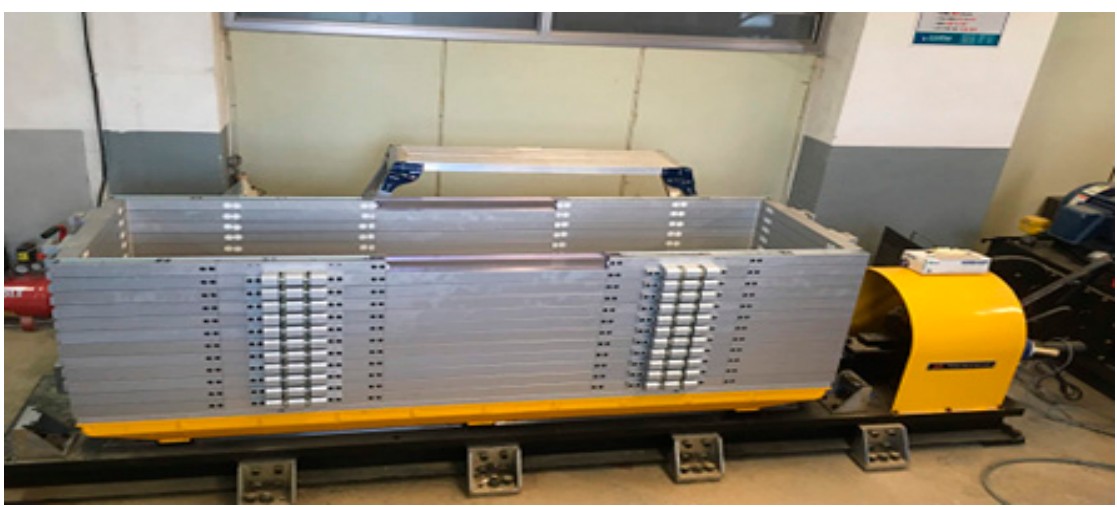

**Figure 2.** Laminar shear box in this study.

### 2.2.3. Accelerometer

The response acceleration measured in this test was expected not to be faster than 20 m/s$^2$ in terms of range, and the assumption was based on the frequency component of response data not being greater than 40 Hz, and the room temperature. The reason why the 20 m/s$^2$ accelerometer was used is that the test level in the 1 *g* shaking table test was less than 20 m/s$^2$. Therefore, the selected accelerometer was ARF-20 A, which can measure up to 20 m/s$^2$. The data logger had 12 channels and was compatible with ARF-20 A as a four gauge sensor, and the data collection interval was maximum 0.001. Figure 3 shows the locations of the accelerometers installed in flat ground and slope model.

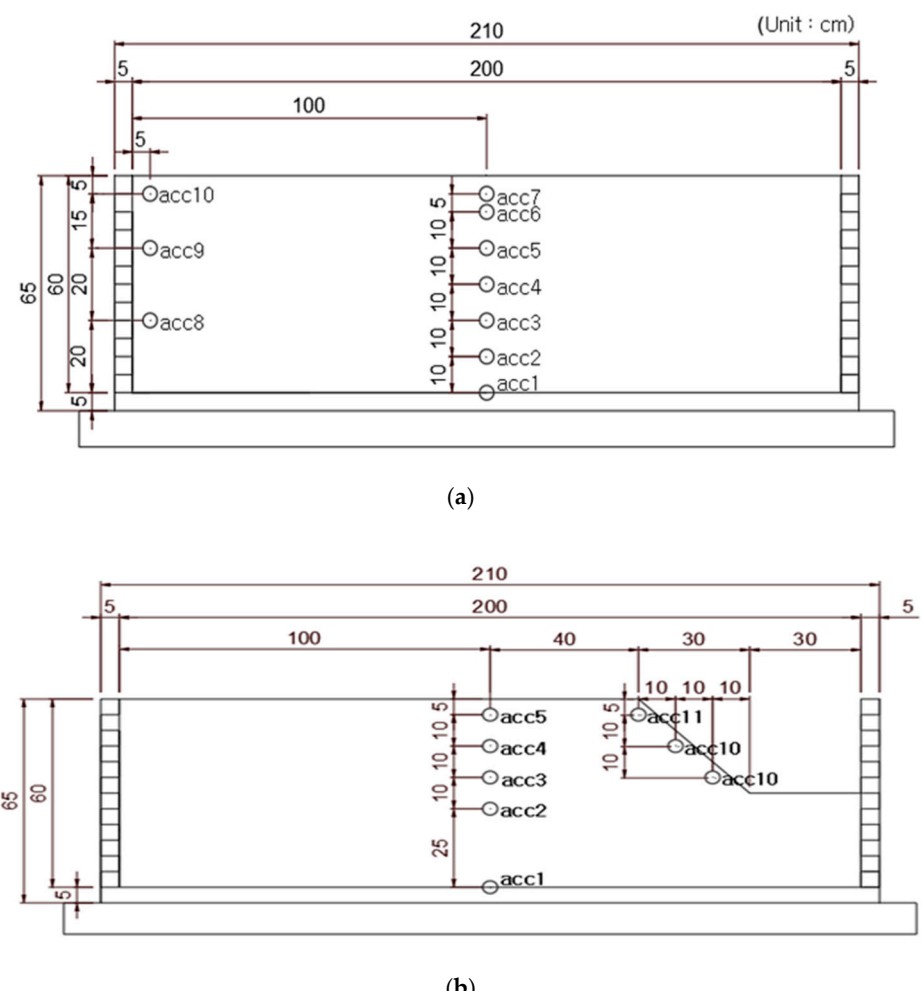

**Figure 3.** Accelerometers setting in the ground: (**a**) Accelerometers setting in flat model; (**b**) Accelerometers setting in slope model.

### 2.3. Numerical Simulation Method

For site seismic response, the characteristics of ground motion can be represented by the amplitude and frequency spectrum of the ground motion. In the site seismic response analysis, acceleration time-history and response spectrum acceleration are important parameters to characterize ground motion. For numerical analysis, DEEPSOIL (1D ground response analysis software) and ABAQUS (finite element analysis software) were used to analyze and compare different types of seismic input motions. In this study, the seismic input motions used are 10 Hz with peak ground acceleration (PGA) equal to 0.07 g and 8 Hz with PGA equal to 0.05 g.

#### 2.3.1. DEEPSOIL Software

DEEPSOIL software is used as the time-domain nonlinear method calculation program. DEEPSOIL software is a unified one-dimensional equivalent linear and nonlinear site response analysis program. 1-D nonlinear and equivalent linear analysis can be performed.

#### 2.3.2. ABAQUS Software

ABAQUS software is a powerful finite element program for engineering simulation, which can solve problems ranging from relatively simple linear analysis to many complex nonlinear problems. ABAQUS software has constitutive models that can simulate stress-

strain behavior of soil, and can accurately establish the initial stress state, which has strong applicability to geotechnical engineering.

### 2.3.3. Comparison of DEEPSOIL and ABAQUS Software

Boundary Conditions

DEEPSOIL software is a 1-D infinite half-space program. Because of the limitations of the DEEPSOIL software, modeling is done only in the flat ground model, modeling and boundary settings are shown in Figure 4a. ABAQUS software is a multi-dimensional modeling program. Artificial boundary conditions have a direct impact on the accuracy of numerical simulation of near-field fluctuations, so the study of artificial boundary conditions is of great significance. In order to eliminate the effect of boundary on model analysis results, the two sides of boundary are redesigned infinite boundaries, as shown in Figure 4b,c.

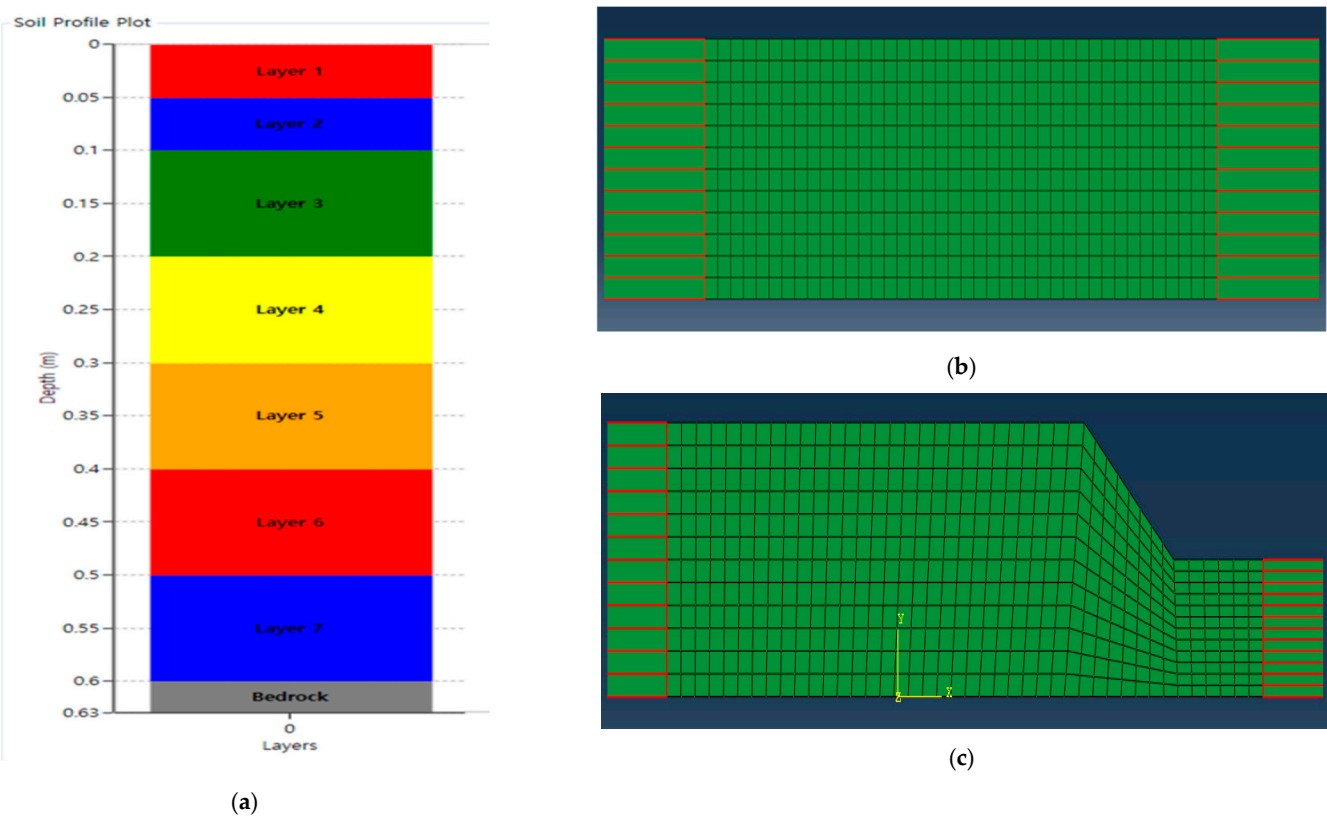

**Figure 4.** Boundary settings: (**a**) 1-D site model in DEEPSOIL; (**b**) Infinite boundary flat model in ABAQUS; (**c**) Infinite boundary slope model in ABAQUS.

Constitutive Models

The DEEPSOIL software uses the Darendeli nonlinear model, while ABAQUS software uses the Mohr-Coulomb linear model. In this study, the main difference between the two software is the constitutive model.

Darendeli [12] proposed a four-parameter model ($\gamma_r$, a, b and $D_{min}$) that can characterize the normalized modulus reduction and material damping curves.

The Mohr-Coulomb model includes five parameters, elasticity modulus E and Poisson's ratio, internal friction angle, cohesion, and dilatancy angle. The Mohr-coulomb model is a first-order approximation result to describe the behavior of rock and soil. It is often used in the preliminary analysis or simplified analysis of the problem. The average stiffness calculated by the Mohr-Coulomb model is constant, and the estimated value of the complex deformation problem can be obtained. Because the Mohr-Coulomb yield

surface has singular points such as spires and corners, the numerical calculation becomes complicated and the convergence is slow in ABAQUS.

Figure 5a,b show the shear stress-shear strain curves of Darendeli model and Mohr-Coulomb model. Figure 5c,d are the $G/G_{max}$-shear strain curve of Darendeli model and Mohr-Coulomb model. In the Darendeli model, $G/G_{max}$ decreases with an increase of shear strain, from linear elasticity to non-linear elasticity to unlimited plasticity. The $G/G_{max}$-shear strain of the Mohr-Coulomb model does not change with an increase of strain. Figure 5e,f show the Darendeli model and the Mohr-Coulomb model using hysteresis curves to calculate the damping ratio. In the Darendeli model, the nonlinearity of shear stress and shear strain will increase energy consumption. Therefore, the damping ratio will increase as shear strain increases. The Mohr-Coulomb model stress-strain curve is linear, so the damping ratio is basically unchanged.

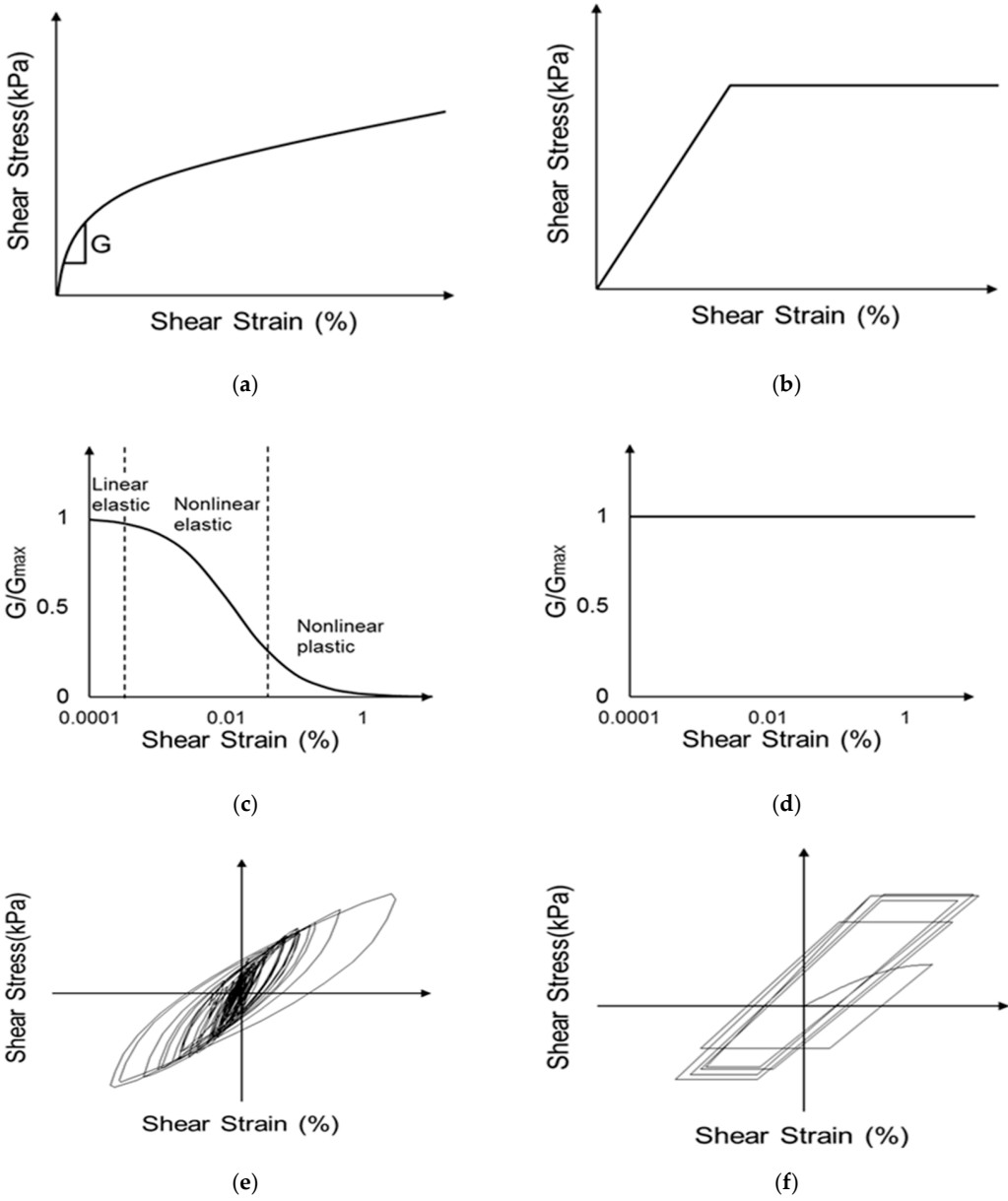

**Figure 5.** Comparison under different models: (**a**) Shear modulus in Darendeli model; (**b**) Shear modulus in Mohr-Coulomb model; (**c**) Normalized shear modulus in Darendeli model; (**d**) Normalized shear modulus in Mohr-Coulomb model; (**e**) Material damping ratio with shearing strain in Darendeli model; (**f**) Material damping ratio with shearing strain in Mohr-Coulomb model.

### 3. Comparison and Analysis of Acceleration Response Results

The behavior characteristics of the model test method were compared with the numerical simulation method. Experiments and analysis values were compared and evaluated using acceleration-time history and spectral acceleration (SA). The experimental values were obtained from accelerometers installed at different depths in the center (or slope) of the laminar shear box of the 1 $g$ shaking table test. The analysis value of DEEPSOIL was obtained at the same depth as the accelerometer buried in the 1 $g$ shaking table test after modeling the semi-infinite ground having the property values of the experiment and performing 1D analysis. In ABAQUS, the ground model used in the shaking table test was modeled in the actual size and the boundary condition was set as the infinite boundary. Then, the analysis value in ABAQUS was obtained at the same location as the location where the accelerometer was buried in the 1 $g$ shaking table test. The response spectrum plots the maximum response to seismic motions with different frequency components, and when the response variable is acceleration, it is called an acceleration response spectrum or spectral acceleration. Spectral acceleration is represented as the maximum response acceleration for a dynamic load with the vibration system of various periods. Using this characteristic, it is possible to confirm the amplification occurring in a frequency period other than the dynamic load used in the experiment. Therefore, using the response spectral acceleration, it is possible to confirm amplification characteristics that are not visible with the sine 8 and10 Hz wave used in this experiment.

*3.1. Comparison of Flat Ground with Normal Elastic Modulus*

3.1.1. Acceleration-Time History

Figure 6 shows the scceleration-time history graph of 1 $g$ shaking table test and numerical analyses using DEEPSOIL and ABAQUS for the same input motion.

As shown in Figure 6, both the 1 $g$ shaking table tests and numerical analyses show the similar acceleration-time history near the bottom. In particular, the 1D ground response analysis results obtained by DEEPSOIL have a good match with the 1 $g$ shaking table test results. This implies that the 1 $g$ shaking table tests with the laminar shear box can simulate the free field behavior of soil subjected to earthquake loading. As shown in Figure 6a–h, it was observed that the time history of the experiment and numerical analysis was slightly different as it goes to the top of the model. This is expected because the more short-period frequency components are added to the overall curve and the amplification difference increases. The short-period frequency components included in the input motion had a minimal effect on the lower part, but the effect on the amplification accumulated toward the upper part of the model is judged to affect the time history graph curve. However, since these short-period frequency (i.e., short period frequency level that is much larger than the natural frequency of the model) components do not coincide with the main amplification period of the model, in other words, no resonance occurs. This effect is unlikely to have a significant effect on spectral acceleration analysis.

3.1.2. Spectral Acceleration

Figure 7 shows the spectral acceleration measured at depths in the 1 $g$ shaking table test and the spectral acceleration obtained from numerical analyses.

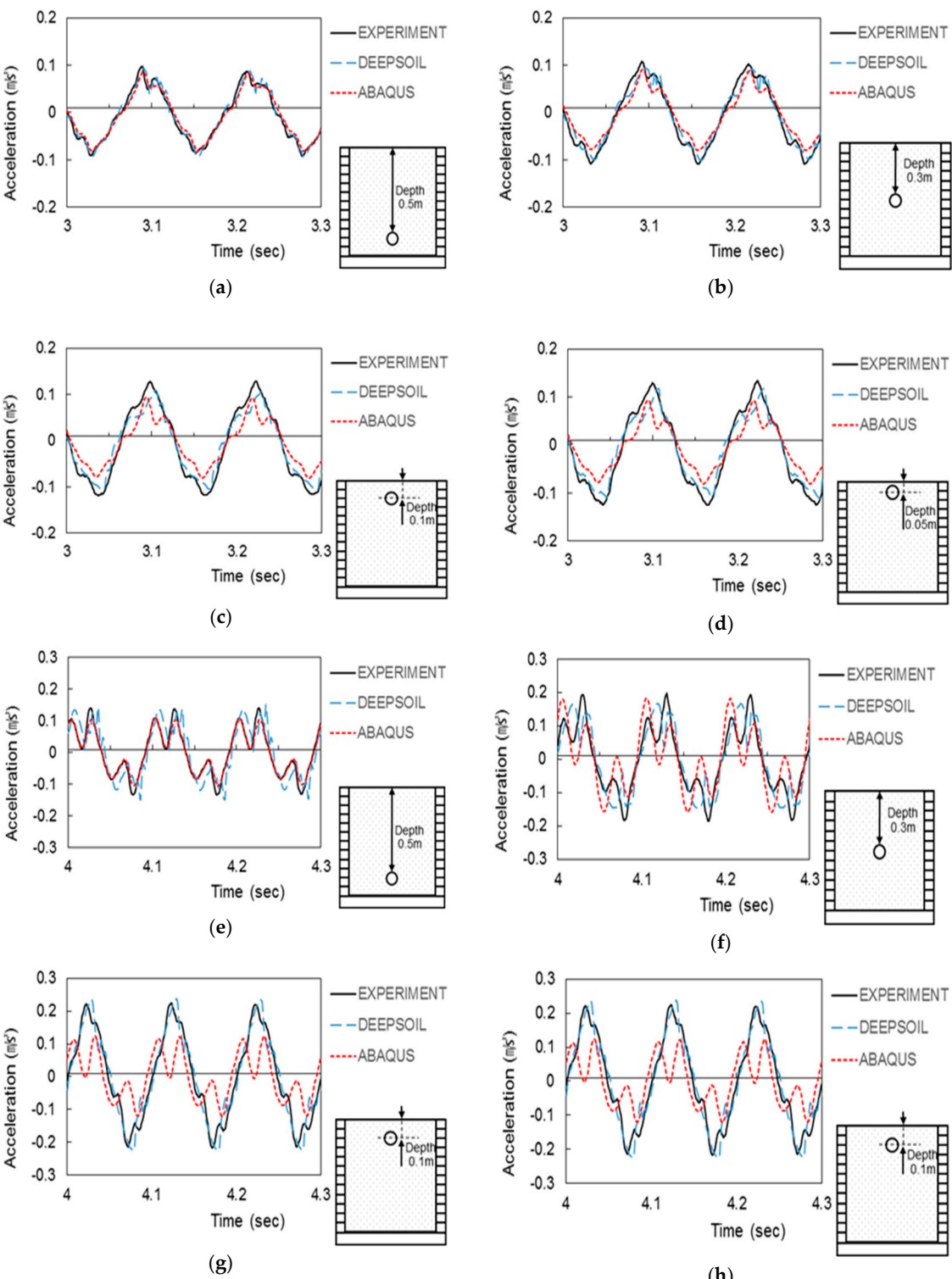

**Figure 6.** Part of acceleration-time history graph for experiment and analysis in flat ground: (**a**–**d**) Sine 8 Hz at depth 0.5 m, 0.3 m, 0.1 m, 0.05 m; (**e**–**h**) Sine 10 Hz at depth 0.5 m, 0.3 m, 0.1 m, 0.05 m.

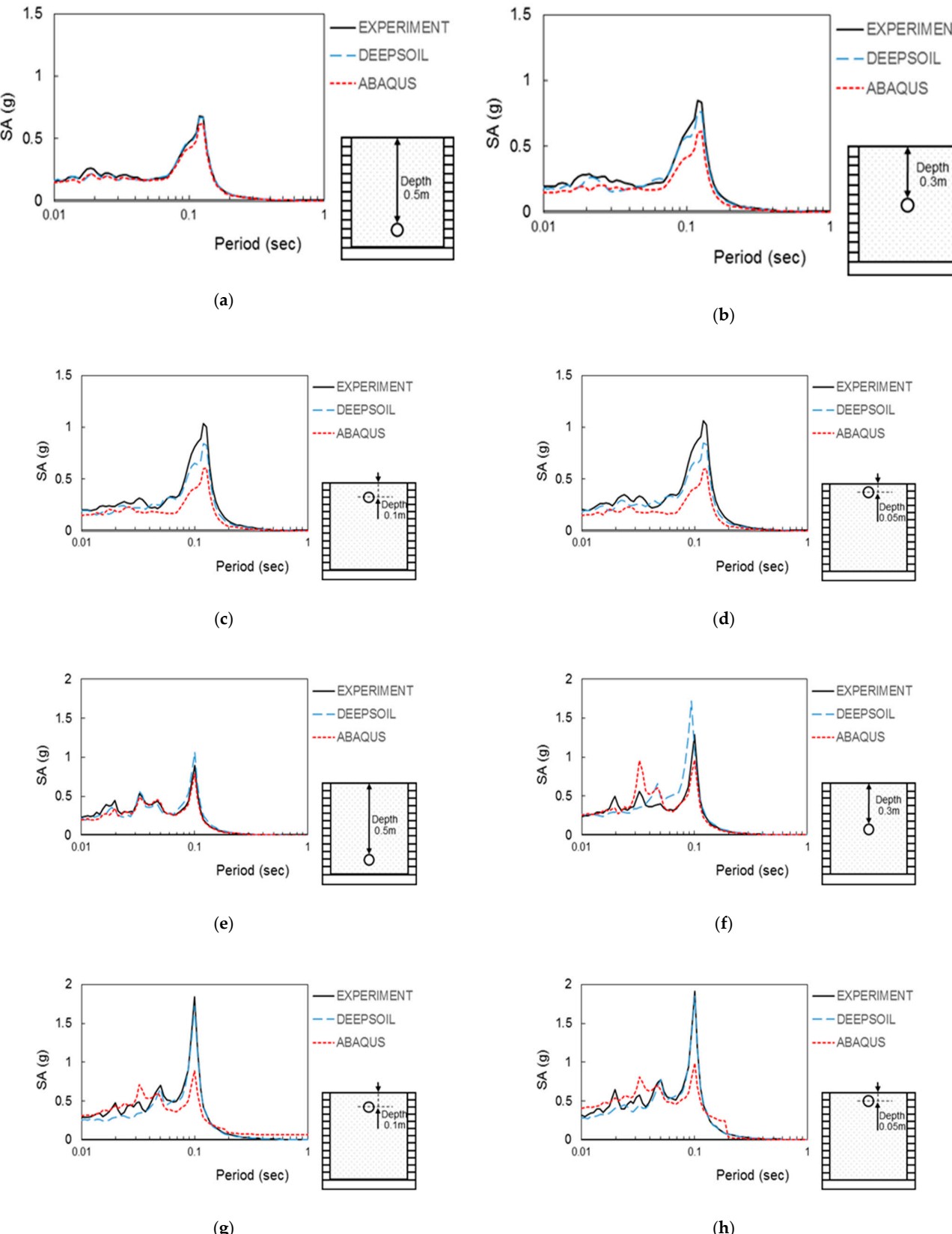

**Figure 7.** Spectral acceleration of experiment and analysis in flat ground: (**a**–**d**) Sine 8 Hz at depth 0.5 m, 0.3 m, 0.1 m, 0.05 m; (**e**–**h**) Sine 10 Hz at depth 0.5 m, 0.3 m, 0.1 m, 0.05 m.

Figure 7a–d present the spectral acceleration graphs of the experiment and numerical analysis for sine 8 Hz. As the depth becomes smaller from 0.5 m to 0.05 m, the greater the difference occurs between the experiment and the numerical analysis. As the depth becomes smaller, the spectral acceleration of numerical analysis is smaller than that of experiment. Figure 7e–h display the spectral acceleration graphs of the experiment and numerical analysis for sine 10 Hz. Near the bottom of the model, DEEPSOIL amplifies slightly higher than the 1 $g$ shaking table test, and the amplification of ABAQUS is low. In Spectral acceleration at a depth of 0.3 m below the surface, DEEPSOIL predicted the very similar SA with the 1 $g$ shaking table test and ABAQUS amplified somewhat lower. Overall, the amplification of ABAQUS is low, so the gap of spectral acceleration between the shaking table test and analysis increases from bottom to surface. The analysis of DEEPSOIL is very similar with the spectral acceleration graph of the shaking table test at all depths.

### 3.2. Comparison of Flat Ground with Adjusted Elastic Modulus

As can be observed from the results in Figure 7, using the elastic modulus for the flat ground does not offer ideal results for the Spectrum acceleration reflection obtained from the analysis with ABAQUS. In order to improve the analysis with ABAQUS, an adjusted modulus of elasticity is introduced, and the following acceleration- time history and spectrum acceleration curves are obtained.

In general, the soil stiffness or elastic modulus depends on the density of the soil. Typical values of elastic modulus of soil are given in Table 2. This data has been recognized by many researchers, and can be used as a guideline. As mentioned above, the classification of the soil is SW-SM. Therefore, according to Table 2, the selected values are between $1.2 \times 10^7$ Pa and $3 \times 10^7$ Paso the adjusted elastic modulus of $2 \times 10^7$ Pa is used in our analysis.

**Table 2.** Typical values of elastic modulus for granular material [13].

| USCS | Description | Loose (MPa) | Medium (MPa) | Dense (MPa) |
|---|---|---|---|---|
| GW, SW | Gravels/Sand well-graded | 30–80 | 80–160 | 160–320 |
| SP | Sand, uniform | 10–30 | 30–50 | 50–80 |
| GM, SM | Sand/Gravel silly | 7–12 | 12–20 | 20–30 |

### 3.2.1. Acceleration-Time History

Figure 8 shows the acceleration-time history graph of the 1 $g$ shaking table test and numerical analyses using DEEPSOIL and ABAQUS for the same input motion.

As can be observed from Figure 8, the acceleration-time history of the model test and numerical analysis is much closer using the adjusted elastic modulus than using the original elastic modulus. As shown in Figure 8, the experiment is very close to the result of DEEPSOIL. Although the acceleration-time history of ABAQUS shifts slightly as the depth decreases, analysis and experimental data are very close for the acceleration trend curve.

### 3.2.2. Spectral Acceleration

Figure 9 shows the spectral acceleration under measured at depths in the 1 $g$ shaking table test and the spectral acceleration obtained from numerical analysis.

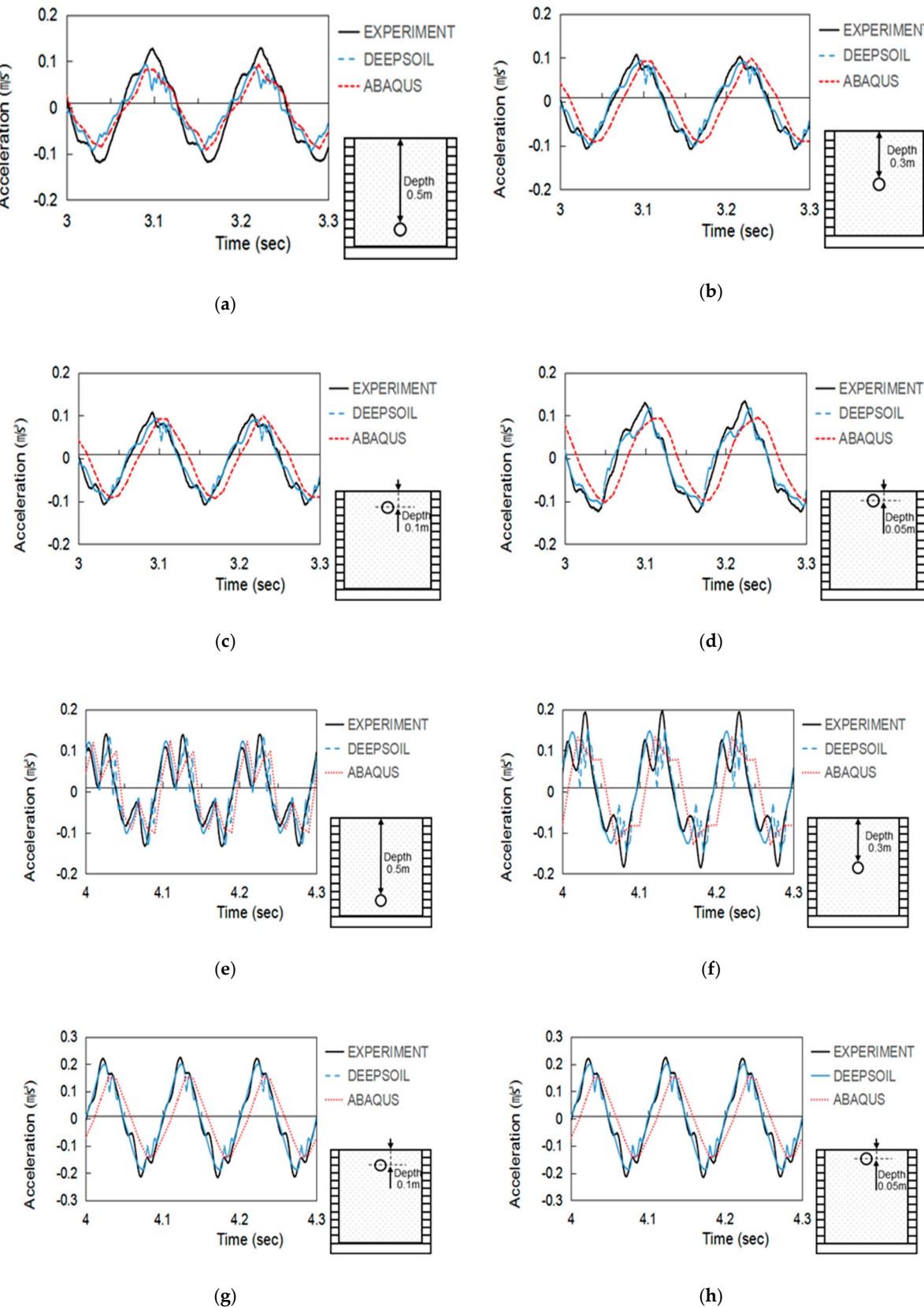

**Figure 8.** Part of acceleration-time history graph for experiment and analysis in flat ground: (**a**–**d**) Sine 8 Hz at depth 0.5 m, 0.3 m, 0.1 m, 0.05 m; (**e**–**h**) Sine 10 Hz at depth 0.5 m, 0.3 m, 0.1 m, 0.05 m.

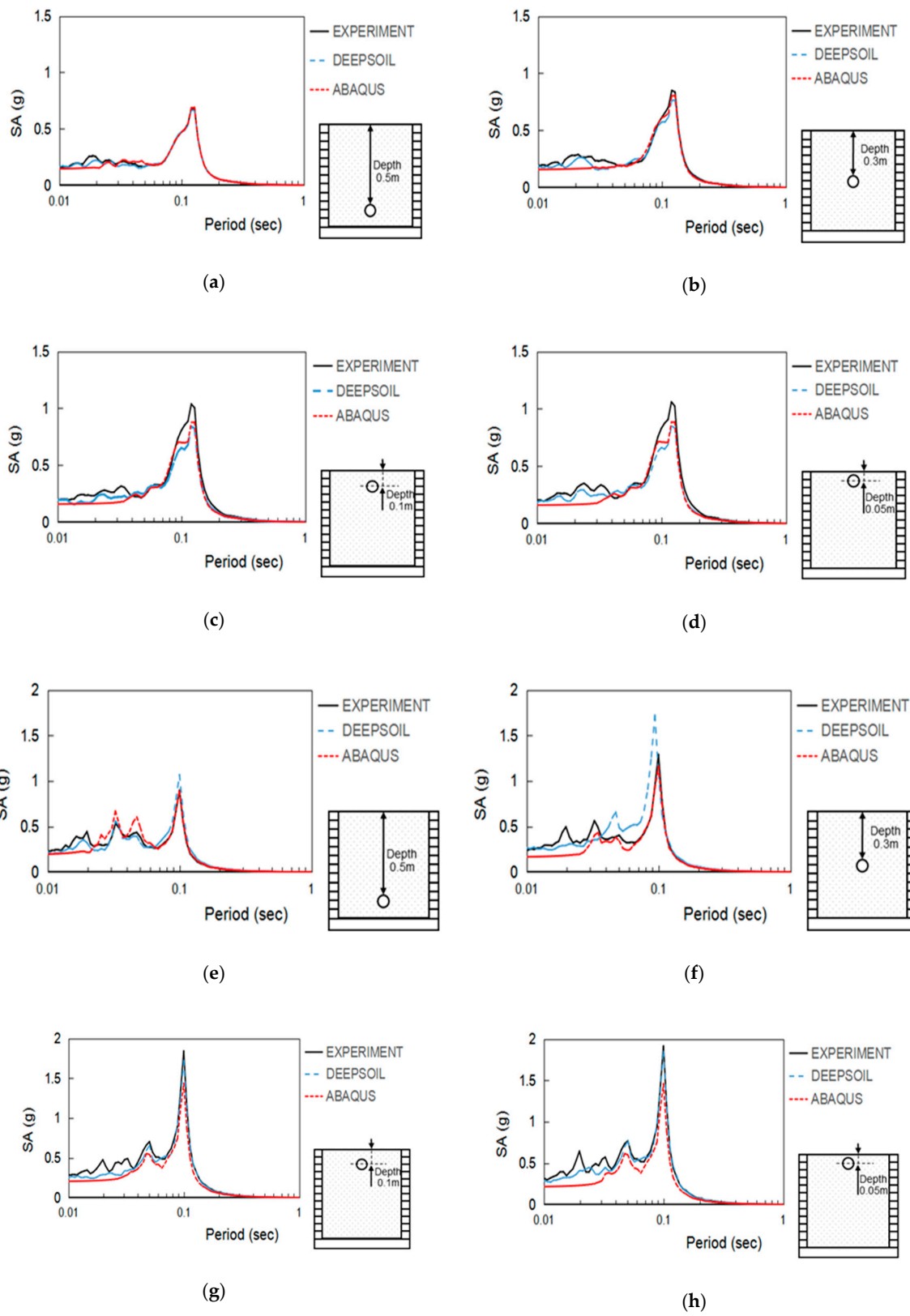

**Figure 9.** Spectral acceleration of experiment and analysis in flat ground: (**a**–**d**) Sine 8 Hz at depth 0.5 m, 0.3 m, 0.1 m, 0.05 m; (**e**–**h**) Sine 10 Hz at depth 0.5 m, 0.3 m, 0.1 m, 0.05 m.

Figure 9a–d exhibit the sine 8 Hz spectral acceleration. As the depth decreases, the difference between experiment and numerical analysis gradually increases. The spectral acceleration amplification obtained from numerical analyses is lower than the experimental spectral acceleration amplification. Figure 9e–h present the sine 10 Hz spectral acceleration. The spectral acceleration of ABAQUS is close to the experimental data at very deep depth, but decreases with depth. The spectral acceleration of DEEPSOIL is closer to that of the experimental data.

Using the adjusted elastic modulus, acceleration time history and spectral acceleration all show very good results for different seismic waves. In this case, the analysis result of ABAQUS is closer to the experimental result than that of DEEPSOIL. As ABAQUS and DEEPSOIL use different constitutive models, they use different parameters. In ABAQUS, direct soil parameter values such as elastic modulus and friction angle are used. In DEEP-SOIL, most of the values such as OMC and PI need to be calculated. So the analysis result of ABAQUS is closer to the experimental result than that of DEEPSOIL. This case verifies the feasibility of the adjusted elastic modulus and the accuracy of ABAQUS modeling.

### 3.3. Comparison of Slope with Adjusted Elastic Modulus

A slope model with a 45-degree inclination angle is used. Accelerometers are set to test the acceleration of the model center and the slope to verify the influence of the slope on the experiment. Because DEEPSOIL is a 1-D semi-infinite space program, the use of DEEPSOIL in this case is not representative, only experimental data and ABAQUS results are used for comparative analysis.

### 3.3.1. Acceleration-Time History

Figure 10 shows the Acceleration-time history graph of the 1 $g$ shaking table test and ABAQUS analysis by sine 8 Hz.

Figure 10a–e are the acceleration-time history at the center of the slope model for sine 8 Hz. As the depth decreases, the acceleration peaks between the experiment and ABAQUS gradually increase, and the difference gradually increases, but the trend is still very close. Figure 10f–h are the slope acceleration time history of the slope model for sine 8 Hz. As the depth decreases, the acceleration peaks of the test and ABAQUS gradually increase, and the acceleration time history is very close.

Figure 11 shows the acceleration-time history graph of the 1 $g$ shaking table test and ABAQUS analysis for sine 10 Hz.

Figure 11a–e show the acceleration-time history at the center of the slope model for sine 10 Hz. As the depth decreases, ABAQUS and the experimental acceleration continue to increase and are very close. Figure 11f–h are the acceleration-time history analysis of the slope model for sine 10 Hz. As the depth increases, ABAQUS and the experimental acceleration continue to increase, and the results are getting closer and closer.

Based on the analysis of acceleration-time history for both flat ground and slope, it turns out that the numerical analyses with ABAQUS can reasonably capture the acceleration-time history obtained from the 1 $g$ shaking table tests.

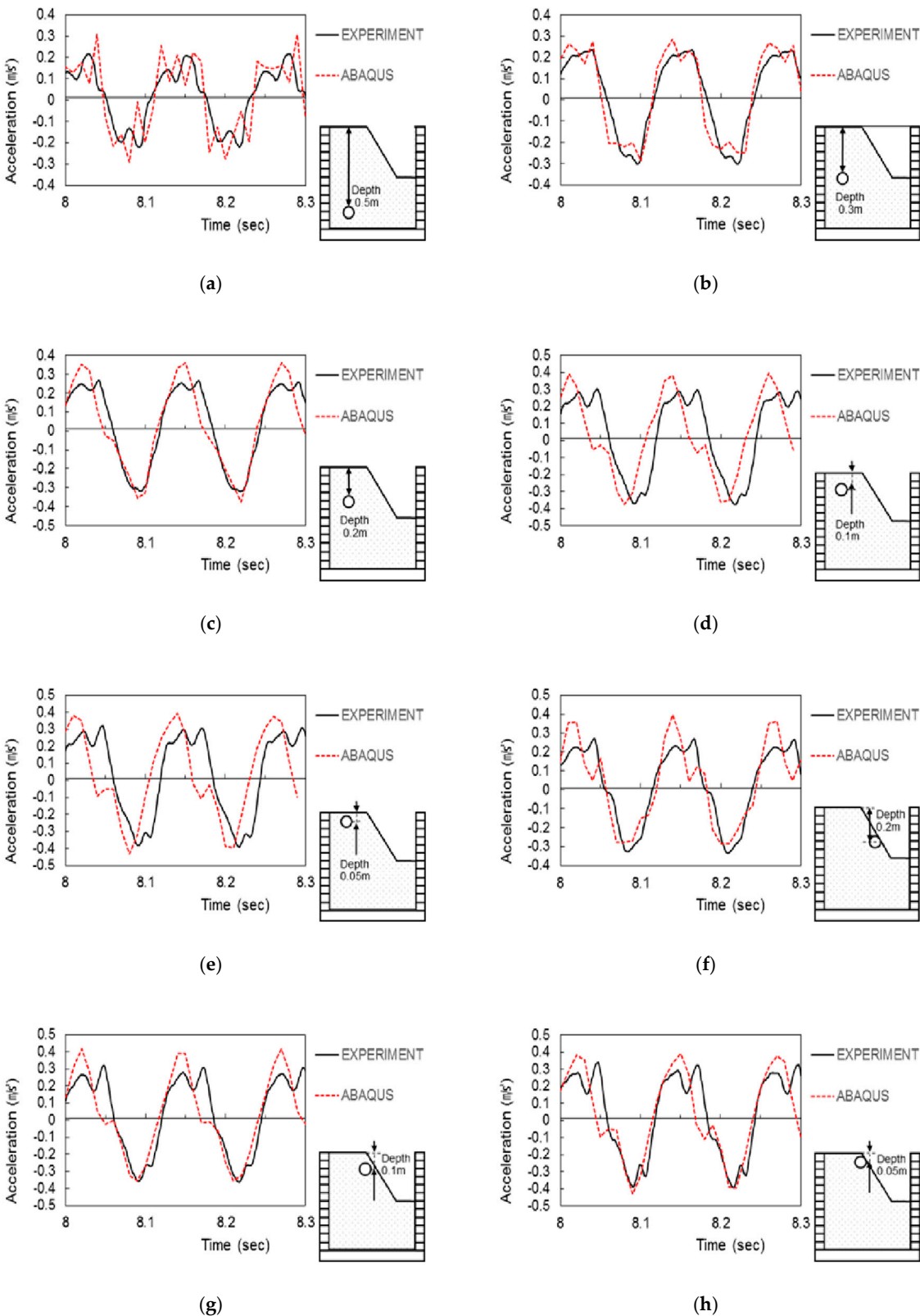

**Figure 10.** Part of acceleration-time history graph for experiment and analysis in slope ground: (**a**–**e**) Sine 8 Hz at depth 0.5 m, 0.3 m, 0.2 m, 0.1 m, 0.05 m; (**f**–**h**) Sine 8 Hz at slope 0.2 m, 0.1 m, 0.05 m.

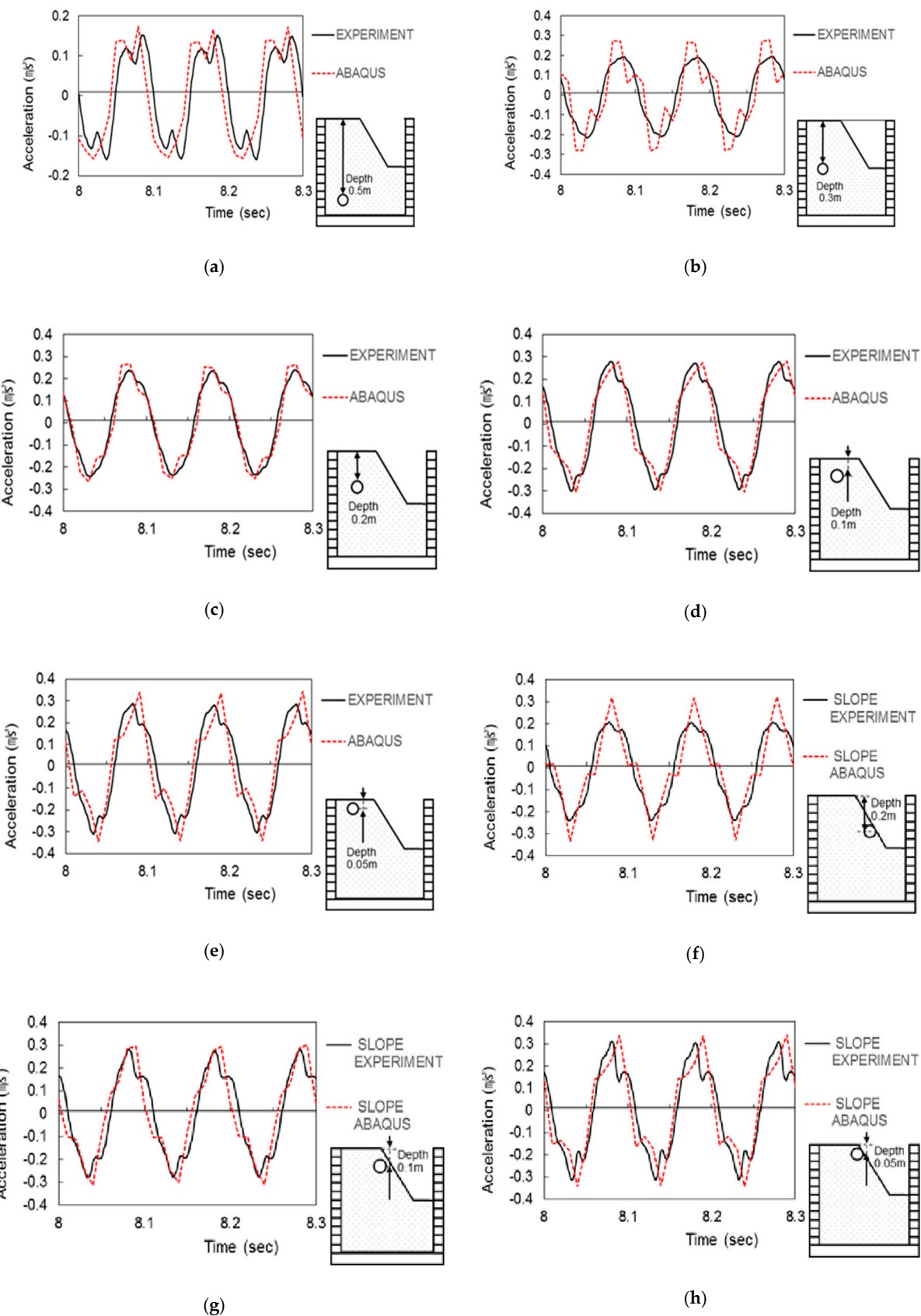

**Figure 11.** Part of acceleration-time history graph for experiment and analysis in slope ground: (**a–e**) Sine 10 Hz at depth 0.5 m, 0.3 m, 0.2 m, 0.1 m, 0.05 m; (**f–h**) Sine 10 Hz at slope 0.2 m, 0.1 m, 0.05 m.

### 3.3.2. Spectral Acceleration

Figure 12 shows the spectral acceleration measured at depths in the 1 *g* shaking table test and the spectral acceleration obtained from ABAQUS.

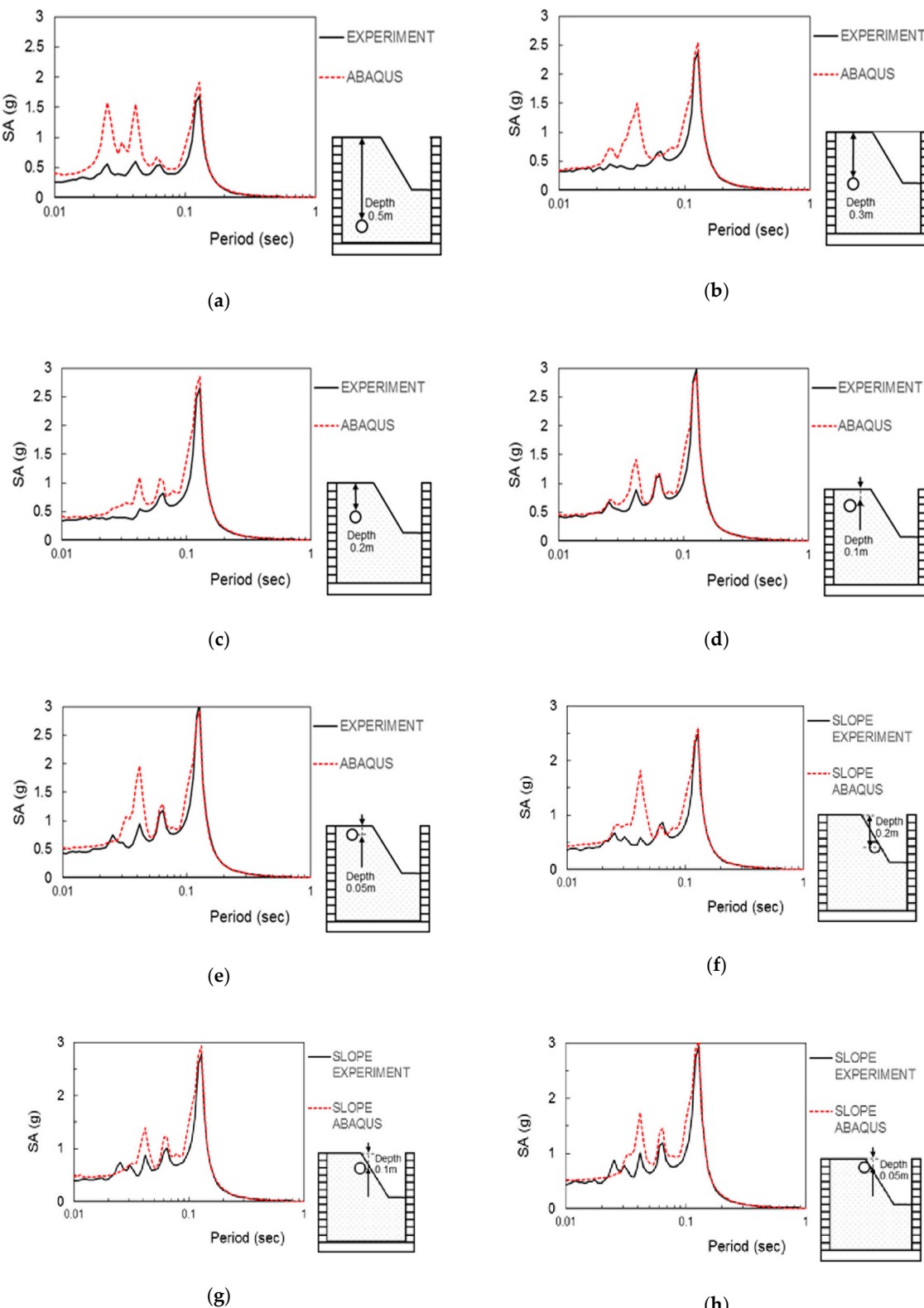

**Figure 12.** Spectral acceleration for experiment and analysis in slope: (**a**–**e**) Sine 8 Hz at depth 0.5 m, 0.3 m, 0.2 m, 0.1 m, 0.05 m; (**f**–**h**) sine 10 Hz at slope 0.2 m, 0.1 m, 0.05 m.

Figure 12a–e present the spectral acceleration of the slope model for sine 8 Hz. As the depth decreases, the trend and peak value of the spectrum acceleration between the experiment and ABAQUS are fairly close. For the left side of the peak, the numerical value of ABAQUS is greater than the experimental value. Figure 12f–h display the slope model spectral acceleration for sine 8 Hz, the general trend and peak value of the experiment and ABAQUS are very close. As the depth decreases, the spectral acceleration value on the left gradually becomes consistent.

Figure 13 shows the apectral acceleration measured at different depths in the 1 $g$ shaking table test and the spectral acceleration obtained from ABAQUS.

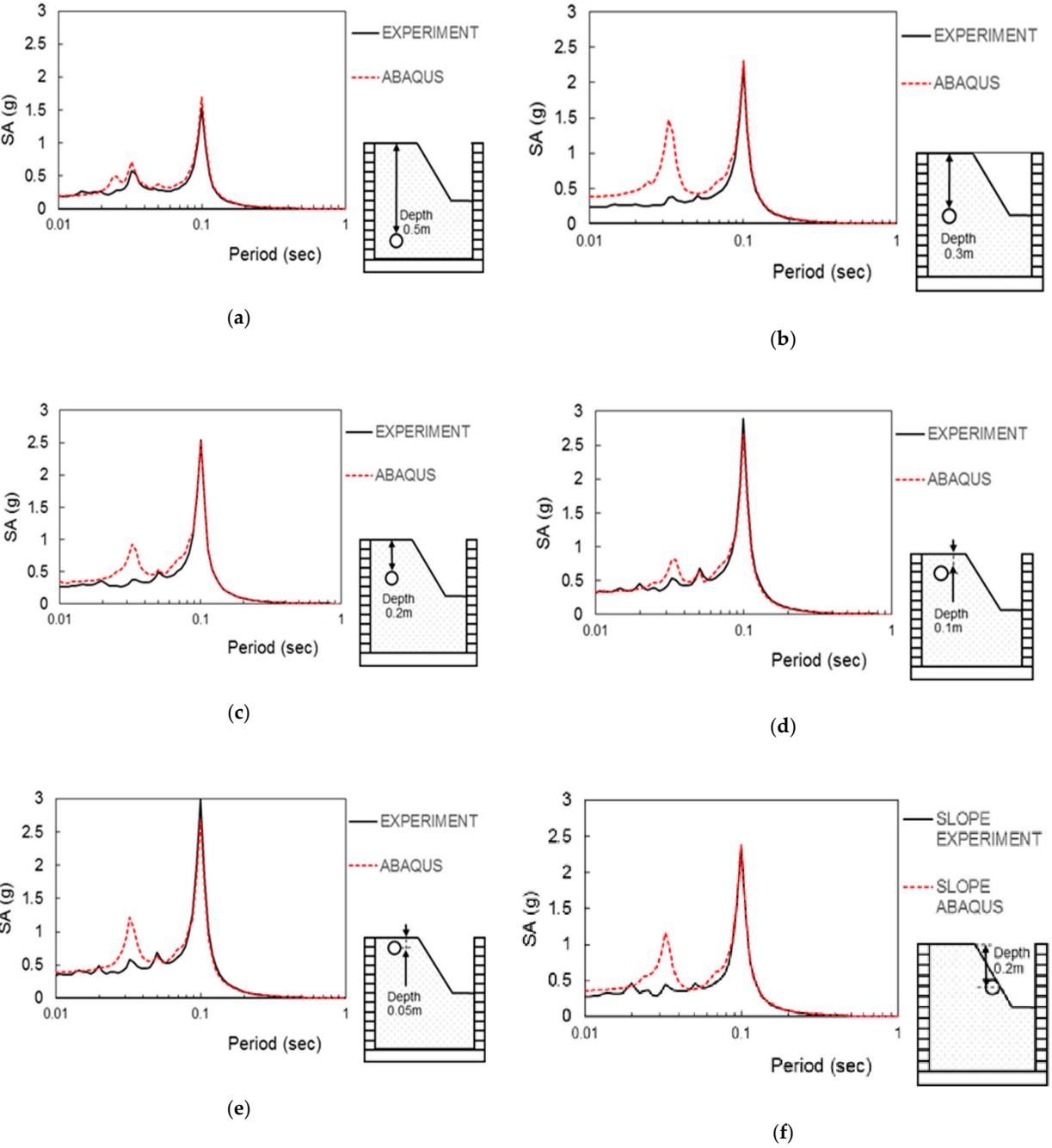

**Figure 13.** *Cont.*

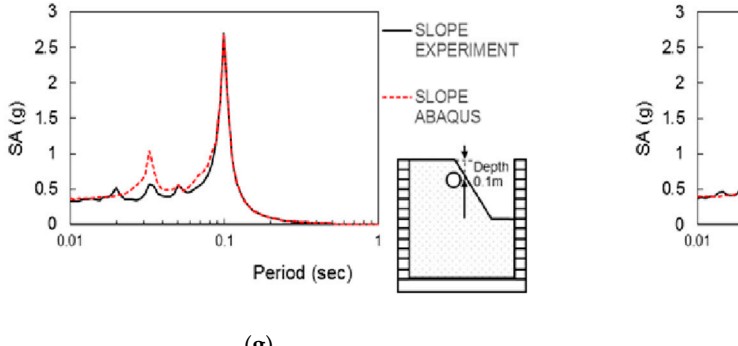
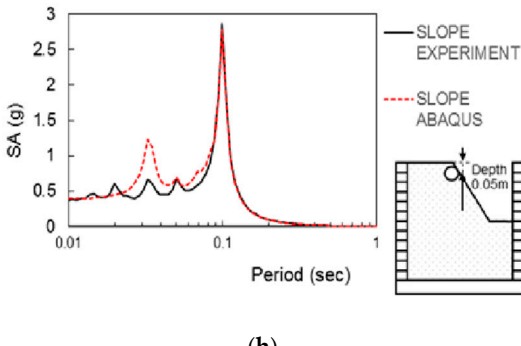

(g)                                                                                                                   (h)

**Figure 13.** Spectral acceleration for experiment and analysis in slope: (**a**–**e**) Sine 10 Hz at depth 0.5 m, 0.3 m, 0.2 m, 0.1 m, 0.05 m; (**f**–**h**) Sine 10 Hz at slope 0.2 m, 0.1 m, 0.05 m.

Figure 13a–e are the spectral acceleration of the slope model under sine 10 Hz. As the depth decreases, the experimental and ABAQUS spectrum acceleration gradually increase. The peak value of the spectral acceleration is always close, and the two tend to coincide. Figure 13f–h are the spectral acceleration of the slope model for sine 10 Hz. The general trend and peak value of the test and ABAQUS are very close. As the depth decreases, the spectral acceleration on the left side of ABAQUS changed little, while the spectral acceleration in the experimental increased, and the gap between the two gradually decreased.

For slope models with input waves of sine 8 Hz and sine 10 Hz, the acceleration -time history and spectral acceleration are very close to the experiment results. Within the allowable range of the elastic modulus, it is possible to find adjustments to the elastic modulus to make the analysis more accurate. In this case, the acceleration magnification difference in the center of the model is not significant, but the magnification difference is obvious on the slope.

## 4. Summary and Conclusions

In this study, the dynamic ground behavior determined through 1 *g* shaking table tests was numerically analyzed using DEEPSOIL and ABAQUS. The conclusions of this study are summarized as follows:

(1) In the ground acceleration profile, the experiment and analysis of DEEPSOIL and ABAQUS showed a consistent tendency to increase as the depth becomes smaller overall. In particular, the analysis of ABAQUS almost coincides with the ground acceleration of the experiment regardless of the input motions.

(2) For the input motion coinciding with the natural frequency of the ground, the spectral accelerations of the experiment and analysis of DEEPSOIL are almost identical and the maximum amplification is also the same. This indicates that using laminar shear box is a good tool to understand the dynamic behavior of soil.

(3) In order to check the amplification effect for the overall frequency period, the spectral acceleration obtained from both DEEPSOIL and ABAQUS are pretty close to that from 1 *g* shaking table test. Therefore, in order to verify the behavior characteristic of ground and dynamic experiment like shaking table test, it would be reasonable to perform different types of analyses and to compare the results comprehensively.

(4) The slope model shows much larger difference in spectral accelerations between experimental and ABAQUS results at the top than that with the flat model. This difference may be due to different boundary conditions and geometry used for the flat and slope model.

(5) In this study, 1 *g* shaking table experiments were performed to simulate flat grounds and slope grounds. The method of experiment and simulation ensures that the data obtained is fairly accurate. For the flat model, the comparison between numerical analysis results with DEEPSOIL and ABAQUS and the experimental results verified the

accuracy of numerical analysis for the flat model modeling. For the slope model, the comparison between numerical analysis results with ABAQUS and experimental results verified the accuracy for the slope model modeling, which provided a good guidance for the future research.

**Author Contributions:** Writing—original draft preparation, Y.J., H.K. (Hoyeon Kim) Review and editing, D.K. Data acquisition H.K. (Hoyeon Kim), Y.L. and H.K. (Haksung Kim) All authors read and agreed to the published version of the manuscript.

**Funding:** This study was supported by Chosun University, 2020.

**Institutional Review Board Statement:** Not applicable.

**Informed Consent Statement:** Not applicable.

**Data Availability Statement:** Not applicable.

**Conflicts of Interest:** The authors declare no conflict of interest.

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
