# Peer review of "Seismic Response of Flat Ground and Slope Models through 1 g Shaking Table Tests and Numerical Analysis"

_applsci, doi:10.3390/app11041875_

Round 1
Reviewer 1 Report
I found the manuscript interesting and informative, and I congratulate you on your efforts. Some of the numerical spectral acceleration results are arguably disappointing, but you obtained good estimates of peak values. I agree that deviations are very likely related to difficulty in replicating experimental boundary conditions. There are a few editorial adjustments required in respect of use of English and the layout, but I am sure those shall be sorted out by you and the journal production staff without my input.
Reviewer 2 Report
The submitted paper discusses about seismic response of flat ground and slope models through 1g shaking table test and numerical analysis. The authors used DEEPSOIL and ABAQUS software to compare the results of flat ground experiments. Then, they used ABAQUS to compare the results of slope experiments. The authors concluded that the numerical analysis results are in good agreement with the experiment results. In my opinion, the paper has been well-organized and can be considered for publication. However, the following comments should be considered by authors before the publication of their manuscript in the journal.
- I am wondering about the novelty of the paper. It should be clearly mentioned in the introduction section.
- In page 4, the authors should explain why the response acceleration measured in this test was not expected to be faster than 20m/s2?
- In page 6, the authors implied that modeling is done only in the flat ground model due to the limitations of the DEEPSOIL software. Does this assumption make a high percentage of error in the results?
- In page 7, the numerical calculation becomes complicated and the convergence is slow in ABAQUS since the Mohr-Coulomb yield surface has singular points. What were the strategy of authors to solve the problem?
- The respected authors implied that since these short-period frequency components do not coincide with the main amplification period of the model, no resonance occurs. The question is that when they observed resonance?
- In page 13, the authors should explain why the analysis result of ABAQUS is closer to the experimental result than that of DEEPSOIL?
- In page 16, is it possible to explain why the numerical value of ABAQUS is greater than the experimental value for the left side of the peak?
- In page 17, why do the spectral acceleration values on the left gradually become consistent with the decrease of depth?
- There are some minor grammatical errors through the paper. Please also take a careful look and revise the quality of the English grammar and syntax where needed.
Round 2
Reviewer 2 Report
I am satisfied with the comments of reviewers. The paper can be published in the current format.